# OpenReview forum: "Node-Centric Knowledge Graph"
_ICLR.cc/2026/Conference — Submitted to ICLR 2026_

### Official Review · Reviewer_zrm3 · 2025-10-31

**Soundness:** 2
**Presentation:** 3
**Contribution:** 2
**Rating:** 4
**Confidence:** 3

**Summary:**

This paper aims to improve the efficiency of updating knowledge graph and to mitigate conflicts from a global perspective, thereby enhancing multi-hop reasoning. The authors propose a node-centric framework, where each node is equipped with a small MLP. This design allows for localized updates—only a small number of parameters need to be modified when new nodes are introduced. Furthermore, the paper employs reinforcement learning  to enable the model to learn strategies for conflict resolution.

**Strengths:**

1. The idea of performing efficient local updates through per-node MLPs, combined with using RL to enhance knowledge graph completion is interesting.

2. The approach maintains a relatively low memory cost, making it potentially scalable to large graphs.

**Weaknesses:**

1. The definition of the output of  MLP_v is not clearly described. It appears to produce a scalar value, but more details on what this output represents and how it is used in reasoning would be helpful.

2. The paper seems to use fixed node features as inputs. It is unclear how relation embeddings are handled and what the input would be if the KG lacks textual features. In most KGC methods, node embeddings are learnable parameters. If textual features are unavailable, node embeddings would need to be trainable—raising the question of whether updates should also include these embeddings, not only the MLP parameters.

3. In Table 1, the baseline models show unusually low performance on the full-graph setting, which differs significantly from results reported in their original papers. Some discussion or explanation would be valuable.

4. It is recommended to include case studies how the proposed method resolves conflicts, since it is not evident whether such conflicts actually exist in the datasets used.

**Questions:**

Please refer to the weakness

---

### Official Review · Reviewer_6puf · 2025-11-01

**Soundness:** 1
**Presentation:** 2
**Contribution:** 1
**Rating:** 2
**Confidence:** 4

**Summary:**

This paper presents a two-tier, node-centric neural knowledge graph (KG) architecture designed to address key challenges in scalability, update efficiency, and synchronization that often arise in traditional KGs. The proposed hierarchical design aims to balance local inference with global consistency.

In the local design, each entity node is equipped with a lightweight Multi-Layer Perceptron (MLP) to capture and reason over its immediate neighborhood relationships. While this idea is conceptually appealing, it may face scalability challenges for very large KGs containing millions of nodes, as maintaining a separate MLP for each node could lead to high computational and memory costs. Exploring shared or parameter-efficient alternatives could make this approach more practical for large-scale applications. At the higher tier, a global control node is introduced to coordinate multi-hop queries and resolve conflicts across local nodes. This hierarchical structure is an intriguing design choice that could potentially improve reasoning efficiency and consistency if implemented effectively.
Regarding the experimental results, there are some points that could be clarified. For example, Table 1 shows several zero values for the WN18RR dataset, which may indicate an implementation or reporting issue. Additionally, the results on the full FB15k-237 and WN18RR datasets appear lower than those typically reported in recent KG completion literature, and a discussion explaining these discrepancies would add valuable context.

Overall, the paper explores an interesting and promising direction by combining local and global reasoning in a hierarchical KG architecture. However, the overall idea would benefit from clearer explanation, and the experimental results appear uncertain and would benefit from additional clarification and validation.

**Strengths:**

1.	This paper proposed a new hierarchical, two-tier architecture for knowledge graphs. The local design uses lightweight Multi-Layer Perceptron (MLP) for each node to capture the its immediate neighborhood relationships, while a global control node is introduced to coordinate multi-hop queries and resolve conflicts across local nodes.

**Weaknesses:**

1.	The results presented in Table 1 appear unusual and may require further verification. For instance, when comparing to other works that use the same datasets, such as KICGPT: Large Language Model with Knowledge in Context for Knowledge Graph Completion, the reported performance differs substantially. On the WN18RR dataset, the performance for 1k samples is reported as 0, and several baseline methods achieve values around 0.001 on the full dataset, which seems unlikely and may indicate an issue with the evaluation or reporting process. Similarly, on FB15k-237, the performance on the full dataset appears notably lower than that on smaller subsets (1k and 5k&10k), which is counterintuitive. A closer review of the experimental setup and evaluation metrics would help clarify these discrepancies and strengthen the reliability of the results.

2.	While the introduction highlights several critical challenges—such as disambiguation, consistency, and efficiency—it remains unclear how the proposed method specifically addresses each of these issues. Providing a clearer explanation or mapping between these identified problems and the corresponding components or mechanisms within the framework would greatly improve the clarity and impact of the paper’s contributions.

3.	In the intro, the third paragraph said “despite these advances” from the previous paragraphs. However, the second paragraph explains some limitations of KGs.

**Questions:**

Please check the weaknesses.

---

### Official Review · Reviewer_y4X4 · 2025-11-02

**Soundness:** 1
**Presentation:** 1
**Contribution:** 1
**Rating:** 0
**Confidence:** 5

**Summary:**

This paper focuses on the problem of knowledge graph (KG) completion (KGC). The authors argue that current methods struggle with three main issues: (1) As KGs grow, some facts become outdated and new knowledge is added. Thus the model needs to be re-trained ; (2) They are unable to deal with entity-specific differences that can cause different entities to be mapped to the same one ; (3) They struggle with multi-hop reasoning. To address these problems, the authors argue that a "node-centric" view of KGC needs to be taken. To this point, they introduce their model, NCC, which learns a separate score function (here a MLP) for each node. The authors argue that this makes it easier to update or add new nodes while only having to update a small portion of parameters. Second, the authors introduce a RL-based solution for (a) resolving disputes between nodes and (b) creating paths for efficient multi-hop reasoning. The authors report results on multiple datasets.

**Strengths:**

1. The authors make a fair point about how new updates can cause issues with trained embedding models. For example, removing or adding a portion of edges does require some retraining.

**Weaknesses:**

1. The biggest issues with this paper is that it omits critical prior work. Specifically, conditional MPNNs [1] such as NBFNet [2], ULTRA [3], and others [4, 5, 6]. All of these methods do not rely on entity embeddings at all and are fully inductive. Regardless, they are still able to outperform embedding-based methods. Furthermore, methods like ULTRA [3] or InGRAM [6] also omit relation embeddings, thus being fully inductive. These models are able to resolve the first two problems introduced in this paper ("Local Disambiguation and Updates" and "Global Consistency"). since no entity embeddings are learnt at all.

2. It's unclear to me how the "Local Disambiguation" issue is a problem is KGC methods. If anything this seems like a problem with KG construction (e.g., confusing the fruit "apple" with the company). I'm not saying this isn't an issue, but I don't see how it applies here at all.

3. The proposed model requires an enormous amount of parameters. For example (I'll omit relations here due to their small number), each entity requires an embedding. The authors say they perform a grid search in the space [32, 64, 128]. Let's assume $d=32$ is the best. Each node then has a two layer MLP, resulting in $d^2$ parameters per MLP. Note that on line 302 the authors argue that each MLP has 16+16=32 hidden units. I have no idea how they arrive at this number. As such, without even considering the RL component, for a KG with $N$ nodes that is $N \cdot d \cdot d^2 = N \cdot d^3$ parameters. It goes without saying that this is a lot. Furthermore, it's easy to see how this can be a problem with entities with few facts (such as WN18RR). Imagine an entity which only appears in one fact in the training graph. It's MLP will barely be trained. As such, this design may heavily disadvantage low degree nodes. This is already a known problem for embedding-based KGC methods [7]. However, it doesn't seem that the authors acknowledge this.

4. It's unclear why the authors use subsets of FB15k-237 and WN18RR given that they aren't big graphs. Furthermore, it's unclear what conclusion we should draw from a result on 1k samples in either datasets versus the full one. The authors consider the full datasets as "large scale datasets" (Line 340). Respectfully, they are clearly not as FB15k-237 is the biggest of the two with about ~272K facts in the training graph. Some good large-scale datasets are those shown in Table 3 in [8].

5. The full dataset results for most baselines is way too low. It's essentially 0 for all methods not NC or NCC. This is clearly much much lower than what is reported in literature. You can look at [2, 4, 5] for examples. The authors don't discuss why this discrepancy exists.

6. The authors argue that existing methods struggle with multi-hop reasoning. However, they don't show that their method actually performs well on multi-hop queries. Showing the overall performance is not enough as it doesn't actually answer their original problem.


[1] Huang, Xingyue, et al. "A theory of link prediction via relational weisfeiler-leman on knowledge graphs." Advances in Neural Information Processing Systems 36 (2023): 19714-19748.
[2] Zhu, Zhaocheng, et al. "Neural bellman-ford networks: A general graph neural network framework for link prediction." Advances in neural information processing systems 34 (2021): 29476-29490.
[3] Galkin, Mikhail, et al. "Towards Foundation Models for Knowledge Graph Reasoning." ICLR, 2024.
[4] Zhu, Zhaocheng, et al. "A* net: A scalable path-based reasoning approach for knowledge graphs." Advances in Neural Information Processing Systems 36 (2023): 59323-59336.
[5] Zhang, Yongqi, et al. "Adaprop: Learning adaptive propagation for graph neural network based knowledge graph reasoning." Proceedings of the 29th ACM SIGKDD conference on knowledge discovery and data mining. 2023.
[6] Lee, Jaejun, Chanyoung Chung, and Joyce Jiyoung Whang. "InGram: Inductive knowledge graph embedding via relation graphs." International conference on machine learning. PMLR, 2023.
[7] Shomer, Harry, et al. "Toward degree bias in embedding-based knowledge graph completion." Proceedings of the ACM Web Conference 2023. 2023.
[8] Ren, Hongyu, et al. "Smore: Knowledge graph completion and multi-hop reasoning in massive knowledge graphs." Proceedings of the 28th ACM SIGKDD conference on knowledge discovery and data mining. 2022.

**Questions:**

1. Why are the baseline results so low on the full datasets?

---

### Meta-Review · Area_Chair_jojk · 2026-01-21

**Summary:**

This paper presents a novel node-centric neural knowledge graph completion (KGC) architecture designed to increase update efficiency and the accuracy of multi-hop reasoning. Reviewers raised concerns about uncited prior work in inductive KGC and about the validity of the experimental results, which reported unusually low baseline results compared to the literature. They also questioned the scalability of the approach and noted that the main claims (faster updates and better multi-step reasoning) were not demonstrated. No rebuttal was offered.

**Reviewer Concerns:**

all remain outstanding, as there is no rebuttal: scalability, empirical proof of faster updates and better multi-step reasoning.

**Reviewer Scores:**

there was no rebuttal, so scores would not change.

---

### Decision · Program_Chairs · 2026-01-26

Reject